# Genomic-Phenomic Reciprocal Illumination: *Desyopone hereon* gen. et sp. nov., an Exceptional Aneuretine-like Fossil Ant from Ethiopian Amber (Hymenoptera: Formicidae: Ponerinae) [note 1]


**DOI:** 10.3390/insects13090796

**Published:** 2022-09-01

**Authors:** Brendon E. Boudinot, Adrian K. Richter, Jörg U. Hammel, Jacek Szwedo, Błażej Bojarski, Vincent Perrichot

**Affiliations:** 1Institut für Zoologie und Evolutionsforschung, Friedrich-Schiller-Universität Jena, Vor dem Neutor 1, 07743 Jena, Germany; 2Institute of Materials Physics, Helmholtz-Zentrum Hereon, Max-Planck-Straße 1, 21502 Geesthacht, Germany; 3Laboratory of Evolutionary Entomology and Museum of Amber Inclusions, Faculty of Biology, University of Gdańsk, 59 Wita Stwosza Street, 80-309 Gdańsk, Poland; 4CNRS, Géosciences Rennes, University Rennes, UMR 6118, 35000 Rennes, France

**Keywords:** Insecta, ant diversification, poneroid clade, Poneria, Miocene, Ethiopia

## Abstract

**Simple Summary:**

We describe a new species of extinct ants from Miocene-aged Ethiopian amber, based on males that resemble species of the relictual lineage Aneuretinae, but which effectively belong to the Ponerinae, as revealed by advanced 3D-imaging technology (synchrotron radiation micro-computed tomography, SR-µ-CT). We subsequently propose a revision of ant classification at the subfamily level. We also recognize that the new species belongs to a new genus based on recent phylogenomic results that have clarified the generic boundaries of Ponerini ants. Our work, therefore, represents an example of reciprocal illumination between phenomic and genomic data.

**Abstract:**

Fossils are critical for understanding the evolutionary diversification, turnover, and morphological disparification of extant lineages. While fossils cannot be sequenced, phenome-scale data may be generated using micro-computed tomography (µ-CT), thus revealing hidden structures and internal anatomy, when preserved. Here, we adduce the male caste of a new fossil ant species from Miocene Ethiopian amber that resembles members of the Aneuretinae, matching the operational definition of the subfamily. Through the use of synchrotron radiation for µ-CT, we critically test the aneuretine-identity hypothesis. Our results indicate that the new fossils do not belong to the Aneuretinae, but rather the Ponerini (Ponerinae). Informed by recent phylogenomic studies, we were able to place the fossils close to the extant genus *Cryptopone* based on logical character analysis, with the two uniquely sharing absence of the subpetiolar process among all ponerine genera. Consequently, we: (1) revise the male-based key to the global ant subfamilies; (2) revise the definitions of Aneuretinae, Ponerinae, Platythyreini, and Ponerini; (3) discuss the evolution of ant mandibles; and (4) describe the fossils as †*Desyopone hereon* **gen. et sp. nov.** Our study highlights the value of males for ant systematics and the tremendous potential of phenomic imaging technologies for the study of ant evolution.

## 1. Introduction

The fossil record of ants is rich, comprising at present 13 out of the 16 extant subfamilies plus six extinct subfamilies, altogether totaling 167 genera and over 760 species [1] from over 65 deposits [2], which span over 100 million years of evolutionary time [3,4]. The accumulation of these valid fossil taxa over cultural time is divisible into two periods marked by a gap of almost two decades. During the Early period (1829–1937), 254 valid species were described at a rate of ~2.3 per year, with multi-year pauses sparsely punctuated by minor descriptions and major monographs (namely, [5,6,7,8,9]). During the Modern period (1956–), 475 valid species were described (excluding the names from Hong [10]) at a rate of ~7.3 per year, with at least one new valid species per year since 1979. Since 2004, exactly 250 valid fossil species have been described, almost equal to the first hundred plus years of represented by the Early period. Critical for the meaningful identification and placement of fossils is the preservation of morphological detail and the systematic concept and definition of applicable taxonomic groups. In this context, the foundational work for contemporary ant classification is the informally Hennigian *Synopsis and Classification* of Bolton [11], which built on Bolton’s *Identification Guide to the Ant Genera of the World* [12], several cladistic and comparative anatomical results of the prior decade, and the Mayr–Emery–Forel–Wheeler–Brown system that came before.

During the past two decades of contemporary myrmecological systematics, the field has been transformed by a wave of Sanger-based phylogenetic studies (e.g., [13,14,15]) followed by a wave of transcriptomic and reduced-enrichment phylogenomic studies (e.g., [16,17]). These genetic and genomic works, informed by the morphological system, have proved to be a catalyst for the reevaluation of evolutionary and taxonomic patterns of ant phenotypes across the phylogeny, from the species to the family level. Although fossils are critical for understanding the diversification and disparification of extant lineages (e.g., [18,19]), these critical specimens cannot be sequenced, thus their biological information remains challenging to capture meaningfully. In analogy to molecular methodology, the study of phenotype was enhanced since the new millennium by readily accessible microphotography rigs and digital image stacking programs, while the phenomic transformation of the field will soon to be realized via desktop and synchrotron micro-computed tomography (µ-CT and SR-µ-CT, [20,21,22]; e.g., [23,24,25,26]).

Capitalizing on the potential for phenomic–genomic reciprocal illumination, we describe a new genus and species of ants from Ethiopian amber. These fossils meet the operational morphological definition of the subfamily Aneuretinae used by paleomyrmecologists, namely that the meso- and metatibiae have a single spur each, the petiole has an elongate peduncle, the gaster is unconstricted, the venation is nearly complete, and the jugal lobe is absent. The Aneuretinae are represented today by a single relictual species endemic to Sri Lanka, while fossils have been attributed to the group from Palearctic and Nearctic deposits with a geological span of the Paleocene to the Eocene. Should the new species truly be a representative of this subfamily, it would represent a significant range and time expansion for the lineage, anchoring the fossil record of the Aneuretinae in both the Afrotropics and the Miocene for the first time. To evaluate this hypothesis, which was derived from direct examination using standard light microscopic methods, we scanned the holotype specimen via synchrotron micro-computed tomography (SR-µ-CT) and generated renders of key body regions for comparative analysis. Based on these results, we test the hypothesis that the fossils belong to Aneuretinae, revise the key to global ant subfamilies and certain higher taxa of the Formicidae, and we discuss the definition of Aneuretinae and the evolution of ant mandibles. Our results indicate that the operational criteria for the morphological identification of Aneuretinae are inadequate, and that a systematic revision of the subfamily is necessary. Ultimately, our study highlights the systematic value of males and the utility of advanced imaging technology for informing the evolution of the ants.

## 2. Materials and Methods

### 2.1. Geological Setting

Ethiopian amber is dug in at least five localities of the Amhara Region, on the slopes of the gorges of the Wenchit, Jamma, and Bashilo rivers and their tributaries that incise the northwestern Plateau of Ethiopia: four outcrops are known in the Semien Shewa (or North Shewa) Zone [27] (see Figure 2 therein), and one in the Semien Wollo (or South Wollo) Zone [28]. Data from the local collectors, palynological analysis of the amber-bearing rocks, the chemical characterization of amber, and the organismal content of amber, indicate coeval, similar environmental deposits for these five localities. In the Woll locality [27] (see Figure 2 therein) investigated by VP in 2019, the amber-bearing stratum corresponds to a fine siltstone/mudstone that is interbedded between two basaltic layers. According to the geological map of the Were-Ilu area [29], these deposits can be attributed either to the pre-Oligocene Ashangi basalt or the Oligocene-Miocene Alajae basalt. In all localities, the palynomorphs from the amber-bearing rocks commonly comprised *Striaticolpites catatumbus*, *Verrucatosporites usmensis*, and *Alnipollenites verus*, three species known to be regionally abundant in the Oligocene to Early Miocene, which accords with the Alajae basalt of the Woll locality. No specific Oligocene taxa were observed, suggesting an early Miocene age (16–23 Ma).

### 2.2. Materials and Specimen Preparation

The type material comprises 13 male individuals found preserved as syninclusions in a piece of clear, yellow-greenish amber from the Semien Wollo Zone, and housed in the Museum of Amber Inclusions at the University of Gdańsk (MAIG, coll. M. Buzalski), Poland, under the collection number MAIG 6016. The exact locality is unknown, as is the case for most Ethiopian amber that can mostly be accessed from gemstone dealers. However, it originates from an outcrop near the town of Weldiya (M. Buzalski, pers. comm., 2013), likely on the slopes of the nearby Bashilo river.

The amber piece was ground using an Avalon sw1 mini water grinding machine with a water-cooled diamond grinding plate (2000 grit), then polished on all surfaces for an optimal view of all inclusions (Figure 1) using wet silicon carbide papers of grits P 600 to 4000 on a Buehler MetaServ 3000 polisher.

### 2.3. Microphotography

To reduce the light scattering of the sample, a drop of water was applied to the amber surface of interest and covered with a glass coverslip. The figured microphotographs are digitally stacked composites obtained using Helicon Focus software, from up to 70 source images taken with Leica Application Suite software on a Leica M205 C stereomicroscope equipped with a Leica DMC4500 digital camera. The resulting images were then enhanced using Photoshop CC 2019 software. Additional images of male mandibles from various ant subfamilies are from AntWeb [30].

### 2.4. Synchrotron Micro-Computed Tomography

The amber specimen was studied using synchrotron radiation based micro-computed tomography (SR-µ-CT). Scanning was performed at the Imaging Beamline P05 (IBL) [31,32,33] operated by the Helmholtz-Zentrum Hereon at the storage ring PETRA III (Deutsches Elektronen Synchrotron—DESY, Hamburg, Germany). For imaging, a photon energy of 18 keV and a sample to detector distance of 300 mm was used. Projections were recorded using a 50 MP CMOS camera system with an effective pixel size of 0.46 µm. 4001 projections were recorded for each tomographic scan at equal intervals between 0 and π. Tomographic reconstruction was done by applying a transport-of-intensity phase retrieval and using the filtered back projection algorithm (FBP) implemented in a custom reconstruction pipeline [34] using MATLAB (Math-Works) and the Astra Toolbox [35,36,37]. For further processing, raw projections were binned twice resulting in an effective pixel size of 0.91 µm for the reconstructed volume.

### 2.5. Data Segmentation and Rendering

To ease data processing, the 32 bit tif image sequence was converted to 8 bit bmp files with Fiji [38]; these files were then downsampled by a factor of 2 resulting in an effective voxel size of 1.82 µm. Whole-body volume renders were made using Amira 6.0 software (Visage Imaging GmbH, Berlin, Germany). Subsequently, segmentation was performed with Amira for the head and mandibles. These structures of interest were manually marked on every 10th slice in the region of the mandible and every 40th slice for the remaining head capsule. The segmentation was then semiautomatically completed using Biomedisa [39]. Because the thin mandibles had too little contrast to be recognized by the Biomedisa algorithm, they were manually added by tracing the space surrounding the clearly recognizable empty mandible lumen, filling out the space between well contrasted areas of the mandible. The segmented materials were then exported with the plugin script “multiExport” [40] in Amira 6.1 as Tiff image stacks. For volume rendering of the exported structures, VG-Studio Max 3.4 (Volume Graphics GmbH, Heidelberg, Germany) was used.

### 2.6. Conceptual

The fossils treated in the present study display a combination of developmental characters that are globally unique among the Formicidae, thus we use the “gap” criterion (i.e., the “morphological species concept”) in determining the species status. With respect to morphological terminology, we followed: Boudinot et al. [41] and Richter et al. [26] for the head; Boudinot [42] for the wings and mesosoma, with mesosomal modifications as recommended by Aibekova et al. [43]; Lieberman et al. [44] for the metasoma; and Boudinot [45] for the genitalia. Our classification follows Bolton [1]; for ranks that are not regulated by the ICZN, we implement clade names from Boudinot et al. [24].

### 2.7. Measurements and Indices

The measurements (all in mm) and indices follow [25] and are those generally used for all ant castes as well as male-specific variables (indicated by an asterisk *). The holotype specimen was measured with the 2D measurement tool in Amira on the volume rendered data. The paratypes were measured with a stereomicroscope ocular micrometer. Note that FCW (frontal carinae width) was not measured for any specimens as these structures not developed in the male, and that FWL (fore wing length) was not measured in the holotype as the wings were outside of the field of view during the scan.
BLBody length: the total body length from the anterior margin of the head excluding mandibles to the apex of the abdomen, measured in dorsal view.HLHead length: the length of the head capsule excluding the mandibles; measured in full-face view in a straight line from a line that spans the anteriormost points of the clypeal lobes to the level of a line that spans the posterior corners of the head capsule.HWHead width: the maximum width of the head immediately behind the eyes, measured in full-face view.HWE *Head width, eyes: the maximum width of the head, including the compound eyes.SLScape length: the maximum straight-line length of the scape, excluding the basal constriction or neck that occurs just distad the condylar bulb.ELEye length: in profile, the maximum measurable length of the compound eye.OLL *Ocellus length, lateral: the maximum length of the lateral ocellus, measured in full-face view.OIL *Inter-ocellus length: the minimum distance between the lateral ocelli, measured in full-face view.WLWeber’s length: the diagonal length of the mesosoma in profile, from the angle at which the pronotum meets the cervix to the posterior basal angle of the metapleuron.ML *Mesoscutum length: the maximum length of the mesoscutum, measured in dorsal view.MW *Mesoscutum width: the maximum width of the mesoscutum, measured in dorsal view.FWL *Forewing length: the maximum length of the forewing from the apices of the axillary sclerites to the wing apex.PHPetiole height: the maximum height of petiole (abdominal segment 2), measured in profile view.PLPetiole length: the maximum length of petiole (abdominal segment 2), measured in dorsal view.PWPetiole width: the maximum width of petiole (abdominal segment 2), measured in dorsal view.GLGaster length: the maximum length of gaster (abdominal tergites 3 to 7), measured in dorsal view.GWGaster width: the maximum width of gaster, measured in dorsal view.CICephalic index: HW/HL X 100.HWI *Head width index: HW/HWE X 100.SIScape index: SL/HW X 100.OIOcular index: EL/HL X 100.OCI *Ocellar index: OLL/OIL X 100.EPIEye Position Index: in full-face view, the distance from a horizontal line that spans the anterior clypeal margin to one that spans the anterior margins of the eyes, divided by the distance from a horizontal line that spans the posterior margins of the eyes to one that spans the posterior corners of the head, X 100.MI*Mesoscutum index: MW/ML X 100.PIPetiolar index: PL/PH X 100.

### 2.8. Repositories

The present work builds on observations accumulated across a few dozen collections as outlined in Boudinot et al. [24]. Extensive reference was also made to AntWeb [30]. Repositories of the primary reference material were as follows:

BEBC—Brendon E. Boudinot personal collection, Jena, Germany.

JTLC—John T. Longino personal collection, Salt Lake City, UT, USA.

MAIG—Museum of Amber Inclusions, University of Gdańsk, Poland.

PSWC—Philip S. Ward personal collection, Davis, CA, USA.

UCDC—Bohart Museum of Entomology, University of California, Davis, CA, USA.

## 3. Results


**
*Systematic paleontology*
**


Family Formicidae Latreille, 1809

*Revision to the male-based key to global subfamilies.* Due to the unconstricted condition of abdominal segment IV in †*Desyopone*
*hereon* **gen. et sp. nov.**, in addition to presence of the pleural sulcus and absence of the jugal lobe, it is necessary to revise couplet 8 from the global male-based key to subfamilies from Boudinot [42]. Modifications to this couplet are indicated by *italics*:

8. Abdominal segment IV with cinctus (=constriction) between the pre- and postsclerite **or** jugal lobe present **or** oblique mesopleural sulcus absent or indistinct ***or***
*petiole without tergosternal fusion* … 9 (Ponerinae, Apomyrminae, Amblyoponinae, Proceratiinae, Dorylinae part, †Prionomyrmecini [Myrmeciinae], and Ectatomminae).

–. Abdominal segment IV without a cinctus **and** jugal lobe absent **and** oblique mesopleural sulcus always present ***and***
*petiole with complete tergosternal fusion, i.e., petiolar posttergites and poststernites fused along their lengths* … 19 (“Formicomorpha” *sensu* Bolton [11], i.e., Aneuretinae, Dolichoderinae, Formicinae).

*Note also*: An additional emendation is necessary: in couplet 11, the specimen that the lead author had previously identified as *Dolioponera* (Ponerini, CASENT090028) and used for construction of the key is correctly identified as an aberrant member of Dorylinae; it may represent the unknown male of *Vicinopone*.

Subfamily Aneuretinae Emery, 1913.

Type genus. *Aneuretus* Emery, 1893.

*Diagnosis (all adult castes)*. All adults of Aneuretinae possess the following diagnostic plesiomorphies **(1–6)**: (1) mandibles shovel-shaped (=“triangular”) [plesiomorphy of Poneroformicia, or at least Doryloformicia]; (2) meso- and metatibiae with one spur each [synapomorphy of clade Dolichoderomorpha]; (3) petiole with complete tergosternal fusion [synapomorphy of clade Dolichoderomorpha]; (4) petiole with an elongate anterior peduncle [possible synapomorphy of clade Myrmechoderines]; (5) helcium infraaxial [possible synapomorphy of clade Myrmechoderines]; and (6) abdominal segment IV unconstricted [synapomorphy of clade Dolichoderomorpha]. Males and queens of Aneuretinae share the following diagnostic traits **(7–12)**: (7) near-complete fore wing venation, with only the subdiscal cell open; (8) crossvein 2rs-m furcal to post-furcal, i.e., 2r-sm meeting Rs at or distad 2r-rs; (9) well-prefurcal fore wing crossvein cu-a, i.e., cu-a meeting M+Cu proximad the split of M+Cu by more than one of its own lengths; (10) absence of free M in the hind wing after the juncture of rs-m and Mf1, i.e., the abscissa between Sc+R+Rs and M+Cu linear, without a kink; (11) hind wing anal cell short, its length less than half that of the basal cell; and (12) absence of the hind wing jugal lobe. Males of Aneuretinae have the following diagnostic plesiomorphy **(13)**: (13) genital gonocoxa and gonostylus not strongly differentiated in size, with the dorsal gonocoxal margin continuing more-or-less evenly to that of the gonostylus. Workers and queens of Aneuretinae share the following diagnostic plesiomorphies **(14, 15)**: (14) mandible with biseriate dentition, i.e., with small teeth interspersed between large teeth [synapomorphy of Dolichoderomorpha]; (15) basal and masticatory margins of mandible not marked, i.e., these margins curving into one another, without a distinct angle [synapomorphy of Dolichoderomorpha].

*Remarks*. The operational paleontological definition of Aneuretinae has relied on character states 2, 4, 6, 7 (regardless of subdiscal cell state), and 12. With the explicit recognition of character state 3—which was previously indicated for the “formicomorph subfamilies” by Bolton [11] (p. 16)—it is possible to reject the placement of †*Desyopone*
*hereon* **gen. et sp. nov.** from the Aneuretinae. The condition of helcial axiality is here reinterpreted from Bolton [11] (p. 18), who described the helcium of Aneuretinae as “high on [the] anterior face of abdominal segment III”, which is interpretable as supraaxial *sensu* Keller [46]. Although the helcial tergite of worker *Aneuretus* is dorsoventrally short, it can be seen that the helcium is at the ventralmost position of the sternum, which does not broaden. An axial helcium is confirmed for the Baltic amber taxa †*Paraneuretus* and †*Protaneuretus* as well. Wing venation was observable for *Aneuretus* and †*Paraneuretus*. Finally, we recognize the mandibular character states 14 and 15 as critical for the identification of Aneuretinae. The states of the mandibles have not been previously remarked upon, but along with the conformation of the clypeus (not defined here), they form the *gestalt* of the Aneuretinae and Dolichoderinae, which was likely used by Wheeler [7] to place †*Paraneuretus* and †*Protaneuretus*, although his justifications were not made explicit. Further refinement of the aneuretine diagnosis via comparative phenomics and traditional comparative morphology is highly desirable.

Subfamily Ponerinae Lepeletier de Saint-Fargeau, 1835.

*Type genus*. *Ponera* Latreille, 1804.

*Male diagnosis*. Males of Ponerinae are best identified at the tribal level as there are as yet no clear male-based synapomorphies for the subfamily. Male Ponerinae share the following key diagnostic plesiomorphies: (1) wing venation complete or nearly complete, with at least four closed cells present; (2) petiole without tergosternal fusion; (3) abdominal segment III not petiolated; and (4) abdominal sternum IX without prongs or teeth.

*Remarks*. The diagnosis provided for Ponerinae above and Platythyreini and Ponerini below collectively represent a revision of the global diagnosis for the subfamily of Boudinot [42]. Male Ponerinae have previously been diagnosed for the Malagasy region [47] and Japan [48,49]. No single character has been discovered yet that uniquely identifies all male Ponerinae. Presence of posterolateral processes on the petiolar sternum which contact the outer margins of the helcial tergite, recognized as a ponerine synapomorphy for the female castes [46], are either poorly developed in males or obscured by the petiolar tergite, thus necessitating focused study. Notably, whereas female Ponerinae display a high degree of specialization with respect to mandibular and leg characters, these are universally lacking in the conspecific male.

Tribe Platythyreini Emery, 1901.

Type genus. *Platythyrea* Roger, 1863.

*Male diagnosis*. In addition to the ponerine plesiomorphies, male Platythyreini are distinguished by the following: (1) mandibles worker-like, shovel-shaped (=“triangular”) [plesiomorphy]; (2) face between antennal toruli distinctly raised, such that the antennal toruli are directed relatively laterad, and are situated dorsad a depression which may receive the antenna [synapomorphy, homoplastic in Ponerini]; (3) antennal toruli usually close to or indenting the posterior margin of clypeus, toruli never distant from the posterior margin by more than half of one of their diameters [apomorphy]; (4) meso- and metatibiae with two spurs each [plesiomorphy]; (5) jugal lobe present [plesiomorphy]; (6) helcium not distinctly infraaxial, being axial to more-or-less axial, i.e., at about or just slightly below the midheight of abdominal segment III [plesiomorphy]; (7) abdominal segment IV with cinctus [plesiomorphy]; (8) cuticle pruinose [synapomorphy, homoplastic in Ponerini and some Proceratiinae].

*Remarks*. Because male Platythyreini have never been explicitly diagnosed, we found it necessary to provide a diagnosis in order to confirm the identification of the fossils in question. We observe that the mandibular form, tibial spur count, and cuticular sculpture of male Platythyreini are sufficient for identification at the global scale.

Tribe Ponerini Lepeletier de Saint-Fargeau, 1835.

Type genus. *Ponera* Latreille, 1804.

*Male diagnosis*. In addition to the ponerine plesiomorphies, male Ponerini are distinguished by the following: (1) mandibles vestigial, with an enlarged mandalus, and being variably lobate, spatulate, spiniform, or nub-like (no exceptions known) [synapomorphy, homoplastic among Formicidae]; (2) face between antennal toruli not distinctly raised, thus toruli directed more-or-less dorsally; if the intertorular region is raised, then this region is grooved in appearance due to impression of the supraclypeal area (=frontal triangle) (*Plectroctena* and *Psalidomyrmex* with medial torular arches raised, but not face; *Hagensia*, *Megaponera*, *Ophthalmopone*, *Simopelta*, some *Euponera*, and some *Odontomachus* with intertorular region raised) [plesiomorphy]; (3) antennal toruli usually distant from posterior clypeal margin (some *Brachyponera*, many *Leptogenys*, *Megaponera*, and *Ophthalmopone* have toruli that are close to the clypeal margin) [plesiomorphy]; (4) meso- and metatibiae with two, one, or no spurs; (5) jugal lobe present or absent; (6) helcium distinctly infraaxial, i.e., situated well below the midheight of abdominal segment III (*Simopelta* is an exception due to softening and reduction in size of the metasoma) [synapomorphy, homoplastic among Formicidae]; (7) abdominal segment IV with or without cinctus; (8) cuticle usually not pruinose, being shiny and variably sculptured (*Belonopelta*, *Hagensia*, *Megaponera*, and *Ophthalmopone* are exceptions) [plesiomorphy].

*Remarks*. Definitive infraaxiality in males, i.e., with abdominal tergum III rising high above the petiole, is a strong diagnostic condition for Ponerinae, as this is an infrequent apomorphic condition at the subfamily level. It also occurs in Dolichoderinae and Formicinae, and to some extent in Myrmeciinae and various Myrmicinae. *Discothyrea* (Proceratiinae) may approach infraaxiality, but the third abdominal tergum is low.


**Genus †*Desyopone* gen. nov. Boudinot and Perrichot**


Type species. †*Desyopone hereon* **sp. nov.**, by present designation monotypy.

*ZooBank LSID*: urn:lsid:zoobank.org:act:7228E671-DF5E-44D1-ADE6-62FB6DB34220.

*Etymology*. The genus name is a portmanteau of the traditional ponerine generic suffix, “-pone”, and the acronym for the Deutsches Elektronen-Synchrotron (DESY), whose storage ring and radiation beamline facilities were used to generate the phenomic data that were crucial for the correct identification of the new taxon.

*Diagnosis*. †*Desyopone* has plesiomorphies 1–4 of Ponerinae and is identifiable as Ponerini at minimum due to the vestigial mandibles and infraaxial helcium. †*Desyopone* and *Cryptopone* are uniquely identified among all Ponerinae by: (1) subpetiolar process completely absent, with the poststernite low and nearly flat in profile. The new genus differs from the males of all known *Cryptopone* by the following: (2) head broader than long, excluding the compound eyes (vs. head narrower than long); (3) mandibles elongate (vs. short); (4) mandibles lobate (vs. spiniform); (5) mesospiracular sclerite evidently absent (vs. this sclerite present); (6) meso- and metatibiae with no spur and one spur, respectively (vs. two spurs each); and (7) petiolar peduncle long, about as long as node is tall (vs. peduncle short, considerable shorter than height of node).

*Remarks*. The identity of *Cryptopone* is significantly clarified by the phenomic data from the new species and the phylogenomic revision of Branstetter and Longino [50]. Prior to this work, the diagnostic importance of the absent subpetiolar process was obscured by the inclusion of *Wadeura guianensis* in *Cryptopone*. Now it is clear that the absence of the subpetiolar process is a unique condition among extant Ponerinae that is shared between †*Desyopone*
**gen. nov.** and *Cryptopone*, and thus constitutes a reasonable autapomorphy within the subfamily for the two genera. No known *Cryptopone*, however, matches the diagnostic character combination of †*Desyopone*
**gen. nov.**, with conditions 3–7 being apomorphic. Critically, the elongate peduncle of †*D.*
*hereon* **gen. et sp. nov.** is nearly unique among Ponerinae; this condition is similarly derived in *Harpegnathos* and is approached by *Dinoponera*, some *Odontomachus* (e.g., *O. chelifer*, *O. coquereli*), and *Platythyrea* (although node at middle of segment rather than posterior).


**†*Desyopone hereon* sp. nov. Boudinot and Perrichot**


(Figure 1, Figure 2, Figure 3, Figure 4, Figure 5)

*ZooBank LSID*: urn: lsid:zoobank.org:act:9E345965-6AA1-42D8-A468-187BABBB38D2.

*Etymology*. The specific epithet gratefully recognizes the Helmholtz-Zentrum Hereon, the research center which funds and operates the Imaging Beamline (P05) at DESY, thus making the present work possible.

*Holotype*. Male (m), MAIG 6016 (Figure 1B, Figure 2A,B, Figure 3 and Figure 4), deposited in the Museum of Amber Inclusions at University of Gdańsk, Poland.

*Paratypes*. 12 males (m). Synincluded with holotype in amber piece MAIG 6016 (Figure 1A,C,D, Figure 2C,D and Figure 5).

*Type locality*. Exact locality unknown in the Bashilo river gorge near Weldiya, Semien Wollo Zone, Amhara Region, Ethiopia.

*Type horizon*. A fine siltstone/mudstone of Early Miocene age (16–23 Ma).

*Material*. Holotype and paratypes 1–4 are nearly complete, preserved without apparent distortion but with integument entirely covered by a white, opaque (bacterial?) coat for paratypes 1, 2, 4 (Figure 1A,C and Figure 2A). Paratypes 5–12 mostly complete but variously preserved, more or less distorted by apparent dehydration (e.g., Figure 1D).

*Diagnosis*. †*Desyopone**hereon* is uniquely identifiable among all Ponerinae by the distinctly elongate petiolar peduncle and the enlarged, lobate mandibles.

*Description*. *Measurements*: Holotype (paratypes) (taken from paratypes 1,2,4)—BL 3.11 (3.00–3.50), HW 0.57 (0.50–0.60), HWE 0.70 (0.66–0.70), SL 0.12 (0.10–0.12), EL 0.23, OLL 0.08 (0.08), OIL 0.16 (0.16), WL 1.25 (1.25), ML 0.52 (0.50–0.56), MW 0.49, FWL (2.75), PH 0.26 (0.30), PL ~0.40 (0.40), GL 1.20 (1.00–1.50), GW 0.62 (0.65–0.80), HWI 81 (76–86), SI 21 (20), OCI 50 (50), PI 154 (133).

*Head* (Figure 2, Figure 3, Figure 4): Head capsule ovoid-elliptical in lateral view; in full-face view, posterior head margin broadly and evenly convex to compound eyes; oral region of head, i.e., the malar areas, clypeus, and mouthparts, narrower than distance between compound eyes; postgenal bridge short, about 2/5 the length of the head in full-face view as measured from the postocciput to the hypostoma; malar areas distinctly developed; clypeus medially bulging, laterally depressed. Compound eyes situated almost entirely in anterior half of head; eyes subspherical and relatively small, with their length being about 1/3 head length. Ocelli distant from compound eyes, with the lateral ocelli separated from the compound eyes by slightly more than one compound eye length. Antennal toruli located at about head midlength in full-face view, distinctly posterad the posterior clypeal margin. Antennal scapes short, just barely longer than wide, with their length distinctly < 2 lateral ocellus diameters; scape length slightly more than one pedicel length, but not more than two. Pedicel about 1/3 the length of the first flagellomere. Flagellum narrow and long, with their length greatly exceeding mesosoma length. Mandibles flat and lobate in appearance, without distinct masticatory and basal margins; medial mandibular margin convex, curving more-or-less evenly around apex, which does not have incurvature; mandible length slightly greater than compound eye length. Labrum and paraglossae dangling at rest, both distinctly narrower than the distance between the mandibular bases. Maxillary stipes without transverse ridging.

*Mesosoma* (Figure 2 and Figure 3): Pronotum short and simple but with distinct muscular convexity as seen in lateral and dorsal views; posterad the anteromedian pronotal lobe (“nuchal lobe”), pronotum in the form of a simple arch, without distinct dorsal and ventral surfaces. Propleurae widely emarginate posteromedially, together forming a broad arch for the prosternum. Prosternum with basisternum apparently arcuate anteriorly; prosternal process developed. Mesoscutum somewhat narrow, with the anteroposterior length slightly greater than the lateromedial width. Notauli developed, and Y-shaped; meeting in the posterior half of the mesoscutum. Parapsides developed, although indistinct. Scutoscutellar sulcus broad, with at least five cross-costae. Mesoscutellum simple, convex, longer than broad in full-face view. Oblique mesopleural sulcus developed. Spiracular lobe (ventrad wing insertions) absent. Mesopleural area divided into dorsal and ventral regions; both regions dorsoventrally taller than anteroposteriorly long. Mesosternal and metasternal regions without processes. Propodeum convex, without armature or distinct sculpturation. Propodeal lobes developed, weak.

*Legs*: Mesotibiae with no spur, metatibiae with a single, pectinate spur. Pretarsal claws narrow and simple, without additional teeth. Arolia well-developed but not very large.

*Fore wings* (Figure 2C,D and Figure 5): Costal vein (C) present, complete. Rsf1 diverging from Sc+R+Rs well proximad pterostigma, with Sc+R abscissa about 1/3 pterostigma length. Pterostigma well-developed, long, and narrow, with its length > 5 × its width. Rf distad pterostigma tubular. M+Cu tubular. Mf1 diverging from M+Cu at or slightly distad crossvein cu-a; this abscissa very weakly curved and meeting Rf1 at a distinct oblique angle. Rs+M tubular. Rsf2–3 diverging from Rs+M proximad 1m-cu. Crossvein 1r-rs absent. Crossvein 2r-rs anterior juncture at 2/3 length of pterostigma; this crossvein meeting Rsf proximad crossvein 2rs-m, which is tubular (paratype 4 has 2rs-m duplicated on the right wing, see Figure 2D). Rsf4+ tubular, meeting Rf distally). Mf2 (=abscissa between Rs+M and 1m-cu) short, shorter than 2r-rs and 2rs-m. Mf3 similar in length to but distinctly shorter than Rsf2–3. Mf4+ tubular proximally, becoming nebulous distally, with this occurring at a distance that is about 3 × the length of 2rs-m. Cuf3 (=abscissa of Cu after 1m-cu) joined to 1A posteriorly; 1A is tubular for its entire length. Submarginal cells 1 and 2 similar in size and shape, but with 2 distinctly smaller than 1. Marginal cell 1 long and narrow, with a length that is about 1.5 × pterostigma length. Discal cell 1 subrhomboidal, its length slightly less than 2 × its anteroposterior width. Subdiscal cell closed.

*Hind wings* (Figure 2C and Figure 5): Wing with eight distal hamuli. Jugal lobe absent. C not distinctly developed. R splitting from Sc+R+Rs distad crossvein 1rs-m; Rf incomplete, not reaching anterior wing margin. Rsf tubular for a distance that is about 2 × the length of Sc+R+Rs distad 1rs-m. M+Cu splitting well distad crossvein cu-a. Mf1 meeting rs-m at a broad, oblique angle; Mf developed as a stud distad this juncture. Cuf developed distad the split of M+Cu, but exact condition uncertain. Crossvein cu-a long, slightly longer than the length of Mf1; this crossvein situated proximad split of M+Cu by about twice its length. Anal vein (A) tubular past its juncture with cu-a. Anal cell relatively long; M+Cu proximad cu-a distinctly longer than M+Cu distad cu-a.

*Metasoma* (Figure 2B and Figure 3A,D,E): Petiole nodiform and distinctly pedunculate, albeit without a marked inflection between its anterior portion and the anterior surface of the petiolar node; peduncle about 2/5 petiole length; petiolar node height about 3/5 entire petiole length; node broad and convex; posterior collar well-developed; tergosternal fusion absent, laterotergites present; sternum low and very weakly sinuate in lateral view, without an anteroventral (=subpetiolar) process, nor with a posterior process; posterior sternal margin distinctly notched. Helcium infraaxial (=below abdominal segment III midheight); helcial tergite broad and overlapping sternite laterally. Prora not distinctly developed. Gastral segments homonomous in appearance and gradually decreasing in length from abdominal segment III to VIII. Abdominal segment IV without cinctus (=constriction). Abdominal spiracles IV–VIII obscured by preceding tergites. Abdominal tergum VIII apparently simple. Abdominal sternum IX lobate and somewhat narrow, with a length that is about twice its width. Cerci (=pygostyles) developed.

*Genitalia* (Figure 2B and Figure 3A): Cupula present (only visible in µ-CT cross-sections). Gonopods longer than tall. Gonostyli broad proximally, indistinctly set off from gonocoxa, and lobate in appearance, being narrowly rounded apically. Lateropenites (=digiti) thickened apically, downcurved. Penites (=penisvalvae) apparently longer than tall, curving strongly to their narrowly lobate apices.

*Setation*: Body with two primary hair classes: (1) short pubescence, which covers the head and all segments of the antennae, mesosoma, legs, and metasoma; (2) long hairs, which are sparse on all body regions, but are denser on the metasoma.

## 4. Discussion

Males are the forgotten caste of ants. Their morphological otherness and ergonomic inutility render them both difficult to identify without associated females and less attractive for social and ecological study. However, because of their relative developmental independence from worker- and queen-fated individuals, male ants provide an additional source of systematic and evolutionary information. Here we describe a morphologically exceptional male ant, †*Desyopone*
*hereon* **gen. et sp. nov.** from Ethiopian amber, whose identity was only soluble due to the interplay of modern informatic technologies—a definitive instance of genomic–phenomic reciprocal illumination.

The new fossil species resembles Aneuretinae in form, as seen via standard light microscopy through the amber matrix. Aneuretinae are of considerable biogeographical, paleontological, and phylogenetic interest, as the subfamily is represented by a single species from Sri Lanka, *Aneuretus simoni* Emery, 1893, and is sister to the Dolichoderinae, a diverse and dominant clade. *Aneuretus simoni* is apparently the relictual lineage of a lost fauna: eight fossil genera across North America, Europe, and Asia have been attributed to the subfamily. Collectively, the Aneuretinae are supposed to be defined by an elongate petiolar peduncle, absence of a gastral constriction and retention of both the sting and complete wing venation. With the exception of the sting, the male-based species †*D.*
*hereon* **sp. nov.** meets all of these definitional requirements and more (Figure 2), as they have large mandibles which are neither spiniform, falcate, nor nub-like, they retain the oblique mesopleural sulcus, they lack armament of the eighth abdominal tergum and ninth sternum, and they have lost the jugal lobe and the anterior meso- and metatibial spurs.

Application of non-destructive phenomic technology (SR-µ-CT), however, crucially reveals that the petiole lacks tergosternal fusion (Figure 3), and that, although the mandibles are large and have a broadly curved medial margin, they have a special developmental reduction (Figure 4). Among the descriptions and treatments of the 11 fossil taxa attributed to Aneuretinae, petiolar tergosternal fusion has never been adduced as evidence, nor discussed, with the sole exception of Bolton [11] (p. 16 in diagnosis of the “Formicomorph” subfamilies), a treatment which was focused on extant taxa. This is a glaring lacuna, which necessitates reconceptualization of the Aneuretinae and revisions to a number of taxon diagnoses and the global male-based key to ant subfamilies (see Systematic Paleontology, above). With respect to the mandibles, we observe that †*Desyopone*
*hereon* **gen. et sp. nov.** lacks an incurvature of the mandibular apex, indicating a condition of true mandibular vestigiality, which not only contrasts with Aneuretinae and most Dolichoderinae, but also highlights the underappreciated value of male mandibles.

In the light of recent phenomic [26,41] and phylogenomic [16,17] studies, a systematic comparison of male mandibles across all ant subfamilies, including stem lineages (unpubl. data), indicates new hypotheses for the evolution of the ants and diagnoses of the family-group classification. First, let us set the following male mandibular definitions: (A) “vestigiality” is the condition of having short mandibles that are not worker-like, being linear, spiniform, lobate, spatulate, nub-like; and (B) mandibular “functionality” as having mandibles that are either (a) worker-like but linear or falcate, or (b) worker-like and shovel-shaped (*sensu* [26]), or (c) not worker-like yet which are large-falcate as occurs in doryline army ants [41]. Armed with these definitions, male mandibular functionality is observed to be widespread, but the shovel-shape does not occur in stem groups or Leptanillomorpha and is a minority condition in Poneria. This pattern (Figure 6) strongly suggests that: (1) the ancestral condition of Formicidae was to have male and female mandibles that are similarly shaped yet not shovel-like; (2) derivation of the shovel-shape is a possible synapomorphy of the Poneroformicia; and (3) derivation of vestigiality in male mandibles occurs with some frequency across the three major clades. At a finer scale within the Doryloformicia, vestigiality was likely secondarily derived independently in some Formicinae, various Myrmicinae, and in a highly diagnostic pattern within Dolichoderinae. The evolutionary causes and consequences of male-worker mandibular developmental decoupling is unknown.

The final question for the identity and relation of †*Desyopone*
*hereon* **gen. et sp. nov.** is resolvable through one more round of molecular phylogenetic illumination. Excluding Leptanillinae (see [52]), the postpetiolated subfamilies (see couplet 5 of [42]), and groups with shovel-shaped mandibles (Figure 6), the new taxon would either be an apomyrmine, amblyoponine, martialine, or ponerine. †*Desyopone*
**gen. nov.** does not match any of the derived conditions of Apomyrminae and placement in Martialinae may be rejected by the unfused condition of the petiole, while the infraaxial helcium rejects both prior subfamilies and the Amblyoponinae, leaving only Ponerini. Within Ponerini (see [15] and [50] for the most complete recent phylogenies), †*D.*
*hereon* **sp. nov.** may be placed with confidence owing to the absence of the subpetiolar process, a condition that is newly recognized here as an otherwise unique synapomorphy of *Cryptopone* in the strict sense, i.e., excluding species that were recovered in *Fisheropone* and *Wadeura* [50]. Because †*D.*
*hereon* **sp. nov.** cannot be sequenced, and given the exceptionally derived mandibles and petiolar peduncle, it is unlikely that this fossil would be recovered within the crown clade of *Cryptopone* via formal statistical analysis. For these reasons, we recognize the fossils described here as representing a new yet extinct genus of exceptional, aneuretine-like Ponerinae.

## 5. Conclusions

Ethiopian amber generally follows the pattern of other Miocene insect deposits, with almost exclusively extant arthropod genera fossilized in inclusions. This holds true for ants, with at least 15 extant genera reported to date [25] (Table 1 therein), while †*Desyopone* **gen. nov.** is the first instance of an extinct, new genus that is endemic to Ethiopian amber to date. It is plausible that †*Desyopone* **gen. nov.** may ultimately be discovered alive somewhere in Africa, as male ants are yet largely unknown from the continent. A similar case occurred with *Gracilidris* Wild and Cuezzo, 2006, a dolichoderine first described from a Dominican amber fossil and later found alive in South America [53,54]. Conversely, *Ravavy* Fisher, 2009 was described from Madagascar before being recently found fossilized in Ethiopian amber [25,55].

The present study is a direct instance of genomic–phenomic reciprocal illumination. Moreover, it provides an object lesson in the interpretation of unusual fossils—particularly those with appealing stories, such as the Aneuretinae, for which petiolar tergosternal fusion must be evaluated. Arising from the µ-CT renders of †*Desyopone*
*hereon* **gen. et sp. nov.**, the global male-based key to ant subfamilies is revised, as well as the definitions of Ponerinae, Platythyreini, Ponerini, and *Cryptopone*. The new species displays an extreme lost phenotype, albeit one which is biased toward underappreciation as the taxon is known only from males. Systematic reevaluation of male mandibular morphology, however, strongly suggests that the derivation of shovel-shaped mandibles is a synapomorphy of the Poneroformicia clade, with male mandibles themselves representing a largely untapped source of information at the genus level. We anticipate that synchrotron micro-computed tomography and combined-evidence phylogenetics will transform insect systematics.

## Figures and Tables

**Figure 1 insects-13-00796-f001:**
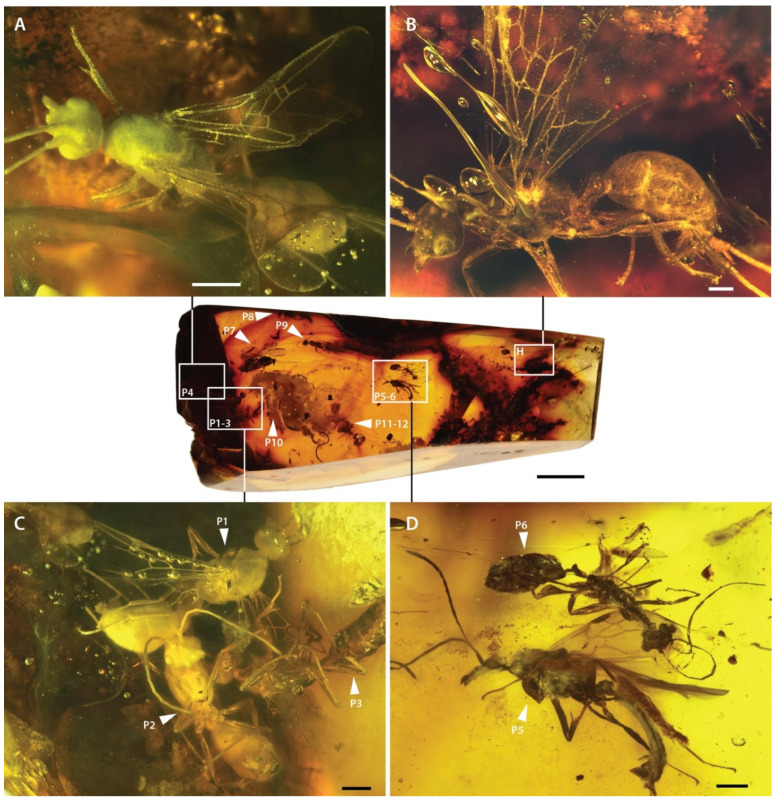
Photograph of entire amber piece MAIG 6016, with indication of type specimens (labeled H for holotype, P1–P12 for paratypes) of †*Desyopone hereon* gen. et sp. nov., and with detailed views of seven of them (**A**–**D**). (**A**) paratype 4; (**B**) holotype; (**C**) paratypes 1–3; (**D**) paratypes 5–6. Scale bars: 0.5 mm.

**Figure 2 insects-13-00796-f002:**
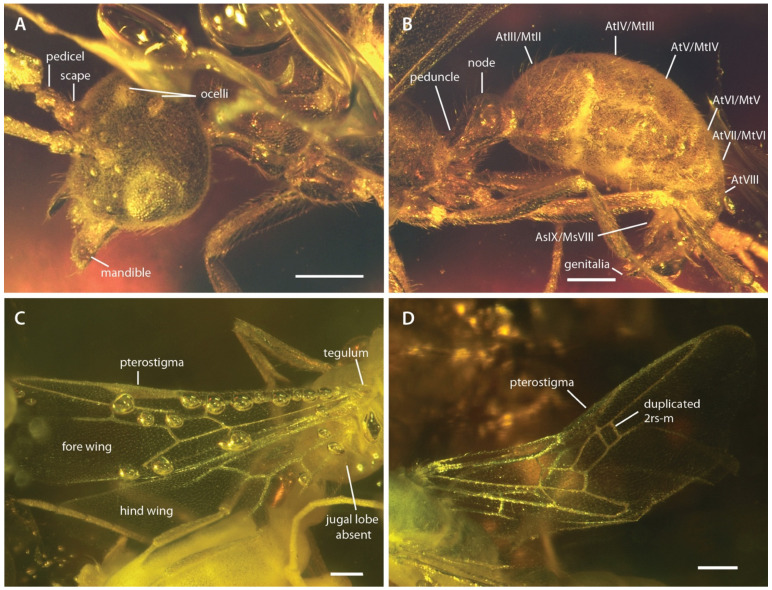
Photographs of †*Desyopone hereon* gen. et sp. nov., MAIG 6016. (**A**,**B**) holotype, anterodorsolateral views of head and metasoma; (**C**) paratype 1, wing view; (**D**) paratype 4, wing view. AtIII/MtII: abdominal tergite III/metasomal tergite II; AsIX/MsVIII: abdominal sternite IX/metasomal sternite VIII. Note that the specimen figured in D has a duplicated crossvein 2rs-m on right fore wing. Scale bars: 0.25 mm.

**Figure 3 insects-13-00796-f003:**
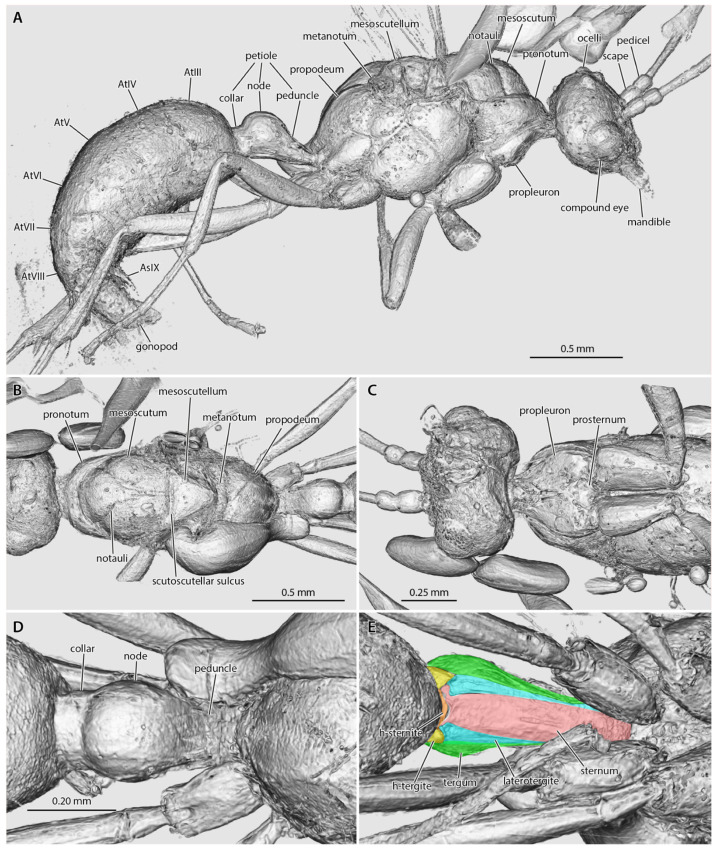
Amira volume renders of the raw data from the †*Desyopone hereon* gen. et sp. nov. holotype MAIG 6016. (**A**) body in lateral view; (**B**) mesosoma in dorsal view; (**C**) head, prothorax, and mesothorax in ventral view; (**D**) petiole in dorsal view; (**E**) petiole in ventral view, highlighting the petiolar tergum, laterotergites, and sternum, and the helcial tergite and sternite.

**Figure 4 insects-13-00796-f004:**
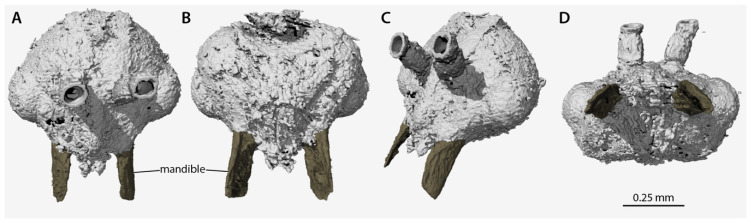
Volume renders of the segmented data from the head of †*Desyopone hereon* gen. et sp. nov., holotype MAIG 6016, showing the highly unusual mandibles. (**A**) full-face view; (**B**) ventral view; (**C**) dorsolateral anterior oblique view; (**D**) oral view. Scale bar approximate due to slightly unequal scaling of images for depiction.

**Figure 5 insects-13-00796-f005:**
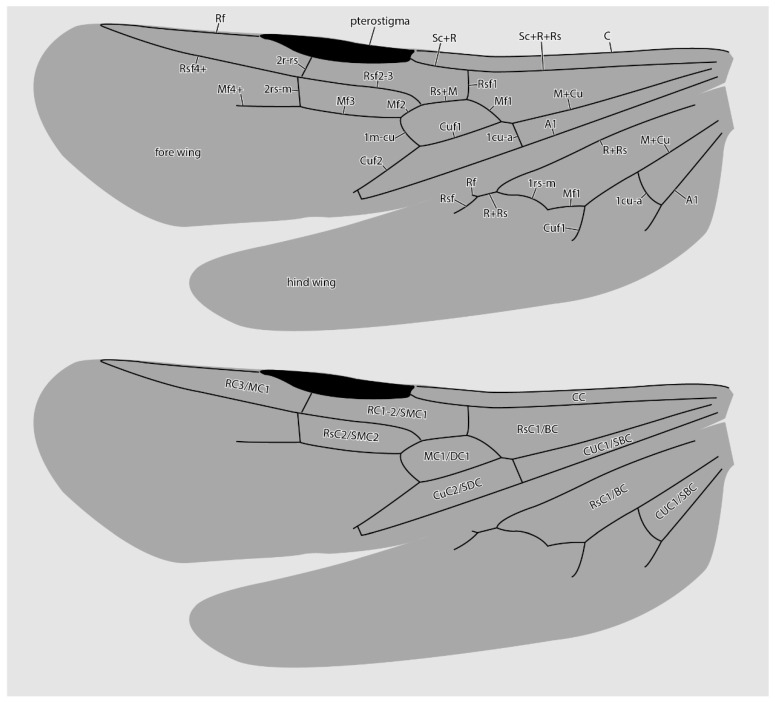
Diagrammatic representations of wing venation (**top**) and cell identities (**bottom**) based on the MAIG 6016 paratype 1 of †*Desyopone hereon* gen. et sp. nov. Cell names: CC = costal cell; RC1-2/SMC1 = first and second radial cells or submarginal cell 1; RC3/MC1 = third radial or first marginal cell; MC1/DC1 = first medial or first discal cell; RsC2/SMC2 = second sectorial or second submarginal cell; CuC1/SBC = first cubital or first subbasal cell; CuC2/SDC = second cubital or first subdiscal cell.

**Figure 6 insects-13-00796-f006:**
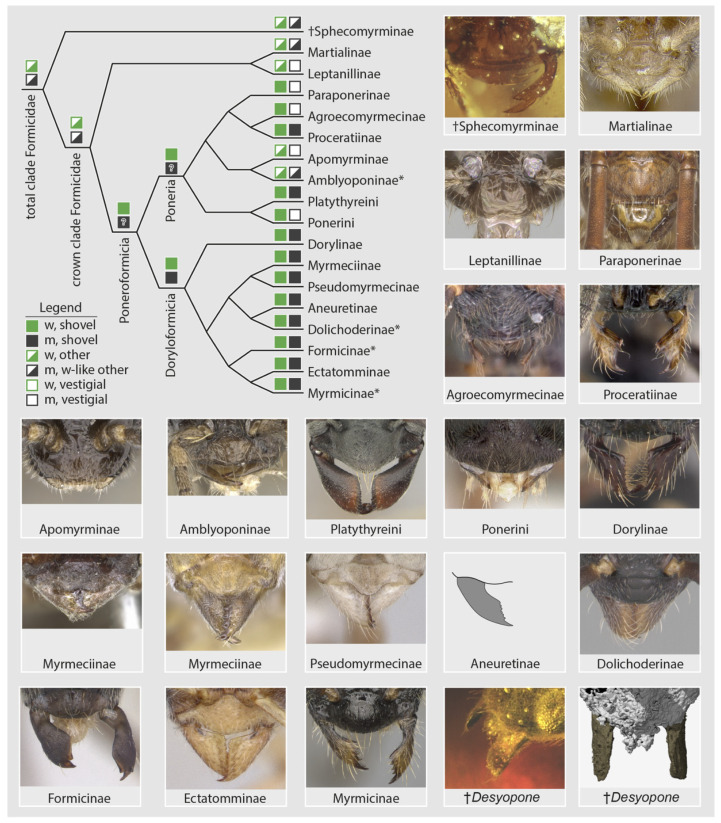
Diagrammatic summary of male mandibular development across the Formicidae at subfamily level. Subfamilies with asterisks (*) have genera or genus groups which are diagnosable by male mandibular vestigiality. In the legend, “w” = worker/female, “m” = male. Images from the top left to the bottom right are from AntWeb [30] with the exception of the Aneuretinae (after [51]) and the †*Desyopone* (this study): †Sphecomyrminae (ANTWEB1032637, J. Chaul), *Martialis* (ANTWEB1041466, B. Boudinot), *Protanilla* th01 (CASENT0119776, M. Esposito), *Paraponera clavata* (CASENT0902407, R. Perry), *Tatuidris* pa01 (CASENT0102681, A. Nobile), *Proceratium* sc02 (CASENT0160796, E. Prado), *Apomyrma* zm01 (CASENT0068418, M. Esposito), *Fulakora* (CASENT0727874, M. Esposito), *Platythyrea lamellose* (CASENT0257315, B. Reynolds), *Pseudoponera stigma* (CASENT0178182, A. Nobile), *Chrysapace sauteri* (CASENT0179567, E. Prado), *Myrmecia auriventris* (CASENT0902789, Z. Lieberman), *Nothomyrmecia macrops* (CASENT0902784, Z. Lieberman), *Pseudomyrmex denticollis* (CASENT0173749, A. Nobile), *Dolichoderus pustulatus* (CASENT0103853, A. Nobile), *Lasius flavus* (CASENT0173150, A. Nobile), *Typhlomyrmex rogenhoferi* (CASENT0006787, A. Nobile), *Myrmica glacialis* (CASENT0862350, A. Nobile).

## Data Availability

The raw scan data will be made available at Zenodo upon acceptance.

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
