# Peer review of "Genomic-Phenomic Reciprocal Illumination: Desyopone hereon gen. et sp. nov., an Exceptional Aneuretine-like Fossil Ant from Ethiopian Amber (Hymenoptera: Formicidae: Ponerinae)†"

_insects, 2022, doi:10.3390/insects13090796_

Round 1

Author Response

The reviewer provided three comments in the Introduction, while the fourth is on the male diagnosis of the Platythyreini.

  1. Lines 76, 77: “I would like to see this "operational definition" [of Aneuretinae] spelled out here.”

In response, we have revised the text as follows (changes marked in purple font):

Capitalizing on the potential for phenomic-genomic reciprocal illumination, we de-scribe a new genus and species of ants from Ethiopian amber. These fossils meet the operational morphological definition of the subfamily Aneuretinae used by paleomyr-mecologists, namely that the meso- and metatibiae have a single spur each, the petiole has an elongate peduncle, the gaster is unconstricted, the venation is nearly complete, and the jugal lobe is absent. The Aneuretinae are represented today by …

  1. Line 86: “This doesn’t read well and confuses a bit. Should be rephrased.”

We have revised the text as follows (changes marked in purple font):

Based on these results, we test the hypothesis that the fossils belong to Aneuretinae, revise the key to global ant subfamilies …

  1. Lines 87, 88: “It should be unpacked a little here, why you found yourselves needing to redefine aneuretines. You should set your readers up more at the begining because it's not apparent until way later in the pub.”

To unpack this, we have added the sentence that is set in purple font:

… and we discuss the definition of Aneuretinae and the evolution of ant mandibles. Our results indicate that the operational criteria for the morphological identification of Aneuretinae are inadequate, and that a systematic revision of the subfamily is necessary. Ultimately, our study highlights the …

  1. Lines 316–332: “It's not clear to me why you're treating platythyreines here. If there is a reason, it needs to be addresses elsewhere in the pub.”

We have updated the remarks subsection of the Platythyreini section as follows (modifications in purple font):

Remarks. Because male Platythyreini have never been explicitly diagnosed, we found it necessary to provide such a diagnosis in order to confirm the identification of the fossils in question. We observe that the mandibular form, tibial spur count, and cuticular sculpture of male Platythyreini are sufficient for identification at the global scale.

Reviewer 2 Report

This is an interesting work that contributes to the study of extinct ants and worth being published. The requirements of the code of zoological nomenclature are observed. Descriptions are correct. All photographs and drawings aspects are well executed. So, I strongly suggest accepting it in this format.

Author Response

We appreciate the positive comments from Reviewer 2. No modifications to the text are necessary.